

# Evaluating Trends and Seasonality in Modeled PM2.5 Concentrations Using Empirical Mode Decomposition

Huiying Luo[1], Marina Astitha[1*], Christian Hogrefe[2], Rohit Mathur[2], S. Trivikrama Rao[1,3]

[1]University of Connecticut, Department of Civil and Environmental Engineering, Storrs-Mansfield, CT, USA
[2]U.S. Environmental Protection Agency, Research Triangle Park, NC, USA
[3]North Carolina State University, Raleigh, NC, USA

*Corresponding author: Marina Astitha, Civil and Environmental Engineering, University of Connecticut, 261 Glenbrook Road, Storrs, CT, 06269-3037, Phone: 860-486-3941, Fax: 860-486-2298, Email: marina.astitha@uconn.edu.

**Abstract.** Regional-scale air quality models are being used for studying the sources, composition, transport, transformation, and deposition of fine particulate matter ($PM_{2.5}$). The availability of decadal air quality simulations provides a unique opportunity to explore sophisticated model evaluation techniques rather than relying solely on traditional operational evaluations. In this study, we propose a new approach for process-based model evaluation of speciated $PM_{2.5}$ using improved Complete Ensemble Empirical Mode Decomposition with Adaptive Noise (improved CEEMDAN) to assess how well version 5.0.2 of the coupled Weather Research and Forecasting model - Community Multiscale Air Quality model (WRF-CMAQ) simulates the time-dependent long-term trend and cyclical variations in the daily average $PM_{2.5}$ and its species, including sulfate ($SO_4$), nitrate ($NO_3$), ammonium ($NH_4$), chloride (Cl) organic carbon (OC) and elemental carbon (EC) . The utility of the proposed approach for model evaluation is demonstrated using $PM_{2.5}$ data at three monitoring locations. At these locations, the model is generally more capable of simulating the rate of change in the long-term trend component than its absolute magnitude. Amplitudes of the sub-seasonal and annual cycles of total $PM_{2.5}$, $SO_4$ and OC are well reproduced. However, the time-dependent phase difference in the annual cycles for total $PM_{2.5}$, OC and EC reveal a phase shift of up to half year, indicating the need for proper temporal allocation of emissions and for updating the treatment of organic aerosols compared to the model version used for this set of simulations. Evaluation of sub-seasonal and inter-annual variations indicates that CMAQ is more capable of replicating the sub-seasonal cycles than inter-annual variations in magnitude and phase.

**Keywords**

Model evaluation, coupled WRF-CMAQ, improved Complete Ensemble Empirical Mode Decomposition (EMD) with Adaptive Noise, Speciated $PM_{2.5}$, Scale Separation, Seasonality, Trend



## 1 Introduction


It is well recognized that inhalable fine particulate matter (PM$_{2.5}$) adversely impacts human health and the
environment. Regional-scale air quality models are being used in health impact studies and decision-making related
to PM$_{2.5}$. Long-term model simulations of PM$_{2.5}$ concentrations using regional air quality models are essential to
identify long-term trends and cyclical variations such as annual cycles in areas larger than what is covered by in-situ
measurements. However, total PM$_{2.5}$ concentrations are challenging to predict because of the dependence on the
contributions from individual PM$_{2.5}$ components, such as sulfates, nitrates, carbonaceous species, and other natural
species. In this context, a detailed process-based evaluation of the simulated speciated PM$_{2.5}$ must be carried out to
ensure acceptable replication of observations so model users can have confidence in using regional air quality models
for policy-making. Furthermore, process based information can be useful for making improvements to the model.
Some of the trend or step change evaluations of regional air quality models in the past have focused on specific pairs
of years (Kang et al., 2013; Zhou et al., 2013; Foley et al., 2015). These studies do not properly account for the sub-
seasonal and inter-annual variations between those specific periods. Trend evaluation is commonly done by linear
regression of indexes such as the annual mean or specific percentiles, assuming linearity and stationarity of time series
(Civerolo et al., 2010; Hogrefe et al., 2011; Banzhaf et al., 2015; Astitha et al., 2017). The problem with the linear
trend evaluation is that there is no guarantee the trend is actually linear during the period of the study because the
underlying processes are in fact nonlinear and nonstationary (Wu et al., 2007).
Seasonal variations are usually studied and evaluated by investigating the monthly or seasonal means (Civerolo et al.,
2010; Banzhaf et al., 2015; Yahya et al., 2016; Henneman et al., 2017). Evaluation of ten-year averaged monthly mean
of PM$_{2.5}$ simulated with WRF/Chem against the Interagency Monitoring of Protected Visual Environments
(IMPROVE) by Yahya et al. (2016) shows that the model captures the observed features of summer peaks in PM$_{2.5}$
with a phase shift of few months. However, according to the analysis (Fig. 10) in Henneman et al. (2017), the
seasonality shown in monthly-averaged PM$_{2.5}$ time series is much less distinguishable compared with that of ozone
and CMAQ (version 5.0.2) does not replicate the monthly PM$_{2.5}$ quite well with large underestimation in the summer
months. In these studies, the seasonality might not be well represented by the preselected averaging window size of
one or three months. In addition, averaging of those monthly or seasonal means across multiple years may conceal the
long-term trends or interannual variations driven by climate change, emission control policies or other slow varying
processes.
To address the above-mentioned problems, we propose a new method for conducting air quality model evaluation for
PM$_{2.5}$ using improved CEEMDAN. Improved CEEMDAN is an Empirical Mode Decomposition (EMD)-based, data-
driven intrinsic mode decomposition technique that can adaptively and recursively decompose a nonlinear and
nonstationary signal into multiple modes called intrinsic mode functions (IMFs) and a residual (trend component)
(Huang et al., 1998; Wu and Huang, 2009; Yeh et al., 2010; Torres et al., 2011; Colominas et al., 2014). It does not
require any preselection of the temporal scales or assumptions of linearity and stationarity for the data, thereby
providing some insights into time series of PM$_{2.5}$ concentrations and its components. Decomposed PM$_{2.5}$ long-term
trend components and annual cycles from observed and simulated PM$_{2.5}$ serve as the intuitive carrier of the trend and





seasonality evaluation. In the meantime, several other IMFs with characteristic time scales ranging from multiple days
to years are also decomposed, enabling model evaluation of the less studied sub-seasonal and inter-annual variations.
Section 2 describes the coupled WRF-CMAQ model simulations and corresponding observations from multiple
speciated $PM_{2.5}$ networks. Section 3 presents an overview of the EMD and improved CEEMDAN technique and the
statistical metrics accompanying model evaluation, including the time-dependent intrinsic correlation (TDIC) on the
decomposed IMFs (Chen et al., 2010; Huang and Schmitt, 2014; Derot et al., 2016). Section 4 describes the findings
on the long-term trend and seasonality in total $PM_{2.5}$ and its components, as resolved by the improved CEEMDAN
technique and includes a discussion on the sub-seasonal, seasonal, and inter-annual variability. The conclusions from
this work are presented in section 5.

## 2 Coupled WRF-CMAQ $PM_{2.5}$ Simulations and Observations

The two-way coupled WRF-CMAQ (version 5.0.2) is configured with a 36 km horizontal grid spacing over the
contiguous United States (CONUS) with 35 vertical layers of varying thickness extending from the surface to 50 mb
(Wong et al., 2012; Gan et al., 2015). Time-varying chemical lateral boundary conditions were derived from the 108
km resolution hemispheric WRF-CMAQ (Mathur et al., 2017) simulation for the 1990-2010 period (Xing et al., 2015).
The simulations are driven by a comprehensive emission dataset which includes the aerosol precursors and primary
particulate matter (Xing et al., 2013, 2015). The readers can refer to Gan et al. (2015) for additional model information
and the trend evaluation against seven pairs of sites from the CASTNET (Clean Air Status and Trend Network) and
IMPROVE networks for 1995-2010. We obtained the 2002-2010 daily average $PM_{2.5}$ and its speciated time series
from the set of simulations with direct aerosol feedback. The earlier years of 1990-2001 are not included in this
evaluation because of the limited availability of speciated $PM_{2.5}$ observations.
To avoid misinterpretation of data due to the presence of missing values, only sites with continuous complete long-
term record for total $PM_{2.5}$ and its speciation including $SO_4$, $NO_3$, $NH_4$, OC, EC, and Cl are studied (Fig. 1). All of the
selected sites have data coverage above 90% each year for at least six consecutive years between 2002 and 2010
(equivalent to 30% for 1-in-3 days sampling sites). This strict data selection led to the sparsity of this type of
observations for the study period. QURE, a rural site carrying out 1-in-3 days sampling of total and speciated $PM_{2.5}$
of $SO_4$, $NO_3$, OC, EC, and Cl, is located in Quabbin Summit, MA. It is one of the three sites from the IMPROVE
network that has at least six continuous years of speciated observations and was selected here to demonstrate the
application of the proposed method in rural areas. It should be noted that the majority of the observed Cl in 2002 and
2003 is negative due to a filter issue problem which was not addressed until 2004 (White, 2008). Thus, simulations of
Cl are only evaluated during 2004-2007 at this site. Station RENO, located in urban Reno, NV, is also a 1-in-3 days
sampling site of total and speciated $PM_{2.5}$ of $SO_4$, $NO_3$, $NH_4$, OC, and EC, and it is the only Chemical Speciation
Network (CSN) site that fulfills this data coverage requirement. The third site ATL in the Southeastern Aerosol
Research and Characterization Study (SEARCH) network is located 4.2 km northwest of downtown Atlanta, GA. It
is the only long-term site available with daily sampling rate (Hansen et al., 2003; Edgerton et al., 2005) that meets the
data coverage requirement. The best-estimate (BE), a calculated concentration intended to represent what is actually



in the atmosphere (Edgerton et al., 2005), of the total PM$_{2.5}$ and SO$_4$, NO$_3$, NH$_4$, and EC components are retrieved for
the evaluation. OC component is a direct measurement. These three sites have a continuous record covering at least 6
years (2002 – 2007 for QURE and ATL and 2002 – 2010 for RENO) that allows an evaluation of long-term trends.

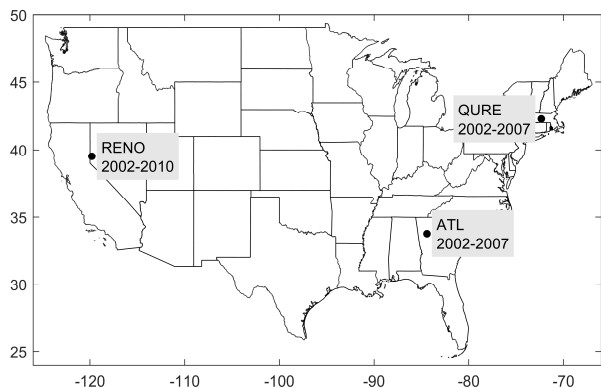


**Fig. 1. Location and data coverage of the PM$_{2.5}$ monitoring sites QURE, RENO and ATL.**
**3 Methodology**
**3.1 Empirical Mode Decomposition**
The Empirical Mode Decomposition (EMD) technique, proposed in the late 1990s, is capable of adaptively and
recursively decomposing a signal into multiple modes called intrinsic mode functions (IMFs), where each mode has
a characteristic frequency, and a residual with at most one extremum (Huang et al., 1998). The decomposed signal
then is expressed as the summation of all IMFs and the residual:
$$x = \sum_{i=1}^{k} d_i + r \qquad (1)$$
where $x$ is the original signal, $d_i$ is the $i$th IMF, $k$ is the number of the IMFs and $r$ is the final residual. Each IMF has
the following properties (Huang et al., 1998):
1) The number of extrema (maxima and minima) and the number of zero-crossings must be equal or differ at most by
one;
2) The local mean at any point, the mean of the envelope defined by local maxima and the envelope defined by local
minima, must be zero.
Nevertheless, "mode mixing" where oscillations with very disparate scales can be present in one mode or vice versa
is commonly reported. To cope with this issue, multiple noise assisted EMD have been developed successively (Wu
and Huang, 2009; Yeh et al., 2010; Torres et al., 2011; Colominas et al., 2014). It is evident that the latest improved
Complete Ensemble EMD with Adaptive Noise (improved CEEMDAN) manages to alleviate the problem of mode
mixing with the benefit of reducing the amount of noise presented and avoiding spurious modes (Colominas et al.,



2014). Moreover, the end effects or boundary effects have been addressed by its predecessor EEMD (Ensemble
Empirical Mode Decomposition) by extrapolating the maxima and minima, and behaved well in numerous time series
with dramatically variant characteristics (Wu and Huang, 2009). The extrapolation of maxima and minima is proven
to be more effective compared with the extrapolation of the signal itself such as repetition or reflection (Rato et al.,

128    2008).

Given the EMD's ability to deal with real-world nonstationary and nonlinear time series data, it is widely used in
engineering, economics, earth and environmental sciences (e.g., Huang et al., 1998; Chang et al., 2003; Yu et al., 2008;
Colominas et al., 2014; Derot et al., 2016). We use the most up-to-date noise-assisted improved CEEMDAN technique
with at least hundreds of noise realizations to decompose observed and simulated $PM_{2.5}$ time series. Readers can refer
to Colominas et al. (2014) for detailed description of the technique and access to the corresponding MATLAB code.
Trial and error attempts are made in setting the input of the improved CEEMDAN function to achieve best mode
separation.
The impact of boundaries on the decomposed annual cycles and the residual is assessed by the variations (standard
deviation) of hypothetical decomposed boundaries by cutting a continuous eighteen-year total $PM_{2.5}$ observation
(North Little Rock, AR) 48 times at different years and times of the year (Fig. S1). The standard deviation is found to
largely diminish within half the annual cycles and could be negligible within one year for the annual cycle. This could
very possibly expand to IMFs with other characteristic scales. Yet, trend components (residuals) show variability
depending on the available time period after cutting. Most of the time, they follow the reference long-term trend
reflected either by the residual or the summation of the residual and the IMF with longest temporal scale decomposed
from the eighteen-year $PM_{2.5}$ (Fig. S1c). This is in line with our expectations as a trend should exist within a given
time span, following the definition in Wu et al. (2007): "The trend is an intrinsically fitted monotonic function or a
function in which there can be at most one extremum within a given data span". Although very strict data completeness
requirement is employed for this study, it should not be conceived as a limitation of the method itself. A sensitivity
test based on a period of nine years of total $PM_{2.5}$ observation at the same site with 99% data coverage shows that even
though variability of annual cycles and long-term trends increases with decreased data availability (100%, 90%,...,
10%), the structure of those components is consistent. The average of 40 realizations of annual cycles and long-term
trend components in each data-completeness scenario is in perfect alignment with that of 100% data completeness
(Fig. S2 and S3). Given the fact that those 40 realizations in each scenario are based on independent random samplings
of the original observations, the increased variability could very possibly result from the difference in the sampled
data itself rather than the method. Thus, the robustness of improved CEEMDAN decomposed annual cycles and long-
term trend is justified. In fact, EMD has been proven to be an effective tool for data gap-filling (Moghtaderi et al.,

155    2012).



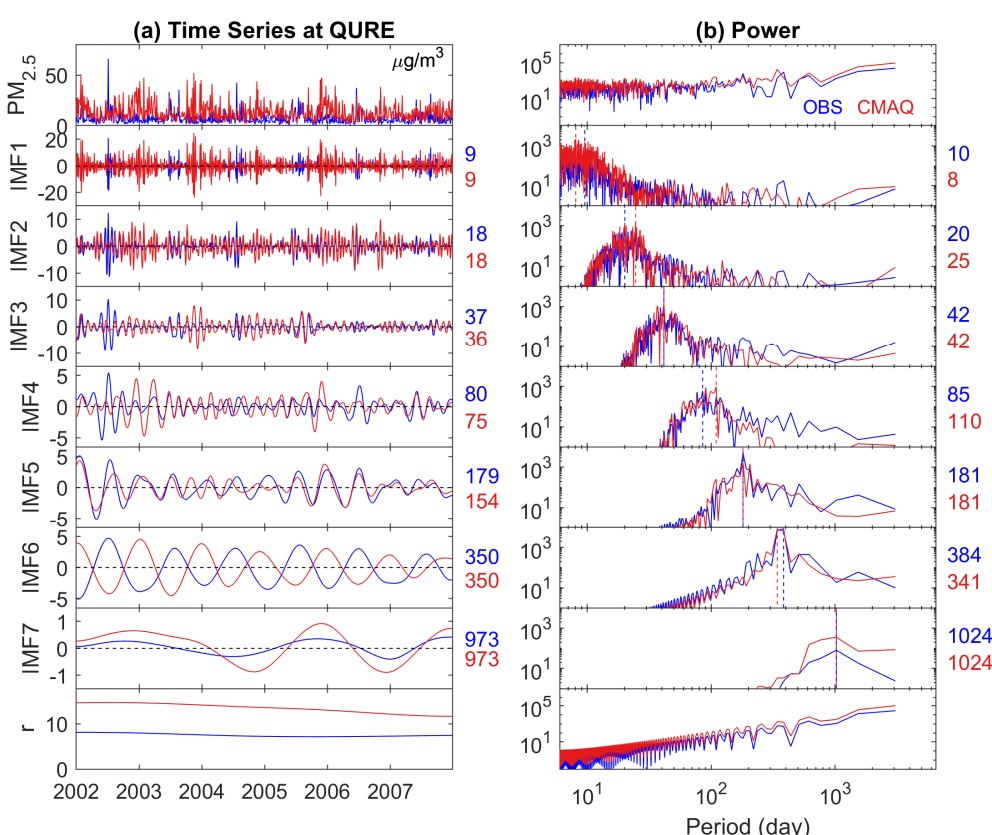

**Fig. 2. Decomposition of observed (blue) and simulated (red) 24-hour average total PM₂.₅ into 7 IMFs and a residual component (trend) at Quabbin Summit, MA using the improved CEEMDAN: (a) Time series of total PM₂.₅, IMFs and the residual component (all with unit of µg/m³); (b) Power spectrum of the corresponding time series. The colored numbers on the right side of time series are the mean period $t_m$ in days, while the ones on the right side of the power spectrum are the peak period $t_p$ in days, which are also indicated by the dashed vertical lines on the power spectrum. Note that the scales for the time series are not all the same. Also, all power spectra are in the log scale and those of the IMFs are zoomed in with a range of $10^0$ to $10^4$ on the y-scale for better visual clarity (compared with $10^{-2}$ to $10^7$ for total PM₂.₅ and the residual component).**

The characteristic period of each IMF can be estimated by the peak period $t_p$ (days) where the power spectrum of the IMF peaks:

$$t_p = \frac{1}{f_p} \qquad (2)$$

in which $f_p$ is the frequency that the power spectrum peaks in the unit of number of cycles per day. The peak estimates can be biased if more than one high-power frequency is located close to each other in one IMF. Thus, power spectrum



is only used as a fast screening tool to determine if a desired decomposition is accomplished. As an alternative
approach, the mean period $t_m$ can be estimated by:
$$t_m = \frac{Time\ span}{(n_{max} + n_{min} + n_{zero})/4} \qquad (3)$$
where $n_{max}$ , $n_{min}$ and $n_{zero}$ are the number of maxima, minima and zero-crossings, respectively, during the
*Time span* (days). As the frequency decreases, the mean period estimates become less accurate because of the limited
time span compared with the length of the cycle and should be carefully interpreted.
An example of the total PM$_{2.5}$ decomposition with improved CEEMDAN at the QURE site shows modes ranging from
very high frequency to very low frequency (IMF1 to IMF7) and a residual (Fig. 2). Mean ($t_m$) and peak ($t_p$)
estimations of the characteristic periods of each IMF are presented on the right side of each mode. Annual cycles and
long-term trend components are well represented by IMF6 and the residual, with the remaining IMFs carrying weekly,
sub-seasonal, seasonal, and inter-annual variations, respectively, for both observed and simulated PM$_{2.5}$ (Fig. 2). We
have noticed that in some rare cases, a spurious mode in the last IMF with synchronous signal and very close scales
to its previous IMF exists. This is possibly due to the fact that the characteristic periods of those IMFs are in proximity
to the span of the studied time span. In these cases, the last two modes are merged by adding those two modes together
to conduct a detailed evaluation as discussed in Section 4.
### 3.2 Statistical metrics
EMD-decomposed IMFs and trend components allow for a detailed time-dependent evaluation of PM$_{2.5}$ and provide
a novel opportunity to trace the performances of specific scales back to the corresponding speciated components. Note
that the trend component is the decomposed residual component from the PM$_{2.5}$ in the unit of µg/m$^3$ and it is not the
traditional concept of trend in concentration per time. In addition to a direct evaluation of its magnitude, we also
calculated its derivative to identify the periods with higher or lower rate of change (concentration per time). Time-
dependent intrinsic correlation (TDIC) is utilized to study the evolvement of the model performance for cyclic
variations throughout time (Chen et al., 2010; Huang and Schmitt, 2014; Derot et al., 2016). It is a set of correlations
calculated for IMFs over a local period of time $I$ centered around time $t$:
$$I(t) = [t - \frac{t_w}{2}, t + \frac{t_w}{2}] \qquad (4)$$
in which $t$ is the center time for the calculation of the correlation and $t_w$ is the moving window length. The minimum
of $t_w$ is set to be the local instantaneous period of the IMF (larger of that in observation or simulation) using the
general zero crossing method to ensure that at least one instantaneous period is included in calculating the local
correlation coefficient (Chen et al., 2010). The maximum of $t_w$ is the entire data period with a traditional overall
correlation being calculated. The empty spaces in the pyramids used to depict the TDIC are an indication that the
correlation is not statistically significantly different from zero. With both decomposed observed and modeled
concentrations in a narrow scale range, the correlation would no longer be contaminated by coexisting signals of
different scales (Chen et al., 2010).



In order to summarize the performance of the decomposed trend component and IMFs, the ratio of the mean
magnitudes of the trend components is defined as:
$$r_{trend} = \frac{Mean_{CMAQ}}{Mean_{observation}} \qquad (5)$$
where $Mean_{CMAQ}$ and $Mean_{observation}$ represent the mean of simulated and observed residual components
respectively. The ratio of the mean amplitude of each IMF is defined by Equation 6, where an example for the annual
cycles is provided:
$$r_{annual} = \frac{RMS_{CMAQ,annual}}{RMS_{observation,annual}} \qquad (6)$$
where $RMS_{observation,annual}$ and $RMS_{CMAQ,annual}$ represent the root mean square of observed and simulated annual
cycles respectively. Finally, the phase shift of an IMF $n$ is defined to be days an IMF decomposed from modeled time
series has to shift in order to achieve the highest correlation ($R_{max}$) with the corresponding IMF with similar scale
from observed PM$_{2.5}$ time series. In practice, $n$ could be as much as a few cycles of the mean period, $t_m$. Here, we
limit the absolute number of shift days to not exceed a half cycle as a reference for the phase shift of an IMF. Thus, $n$
satisfies $-\left(t_m/2\right) \le n \le \left(t_m/2\right)$ with $t_m$ being the larger mean period in observation or simulation. It becomes
$-0.5 \le n/t_m \le 0.5$ in terms of number of cycles.

## 4 Results and Discussion

### 4.1 Temporal scales

Temporal scales in PM$_{2.5}$ resolved by EMD depend solely on the intrinsic properties of the data itself. These properties
include underlying characteristics of specific PM$_{2.5}$ concentrations, the data sampling frequency, which determines the
scales that can be resolved in the high frequency IMFs, and the time span for the data coverage, which could possibly
play an important role in differentiating the low frequency IMFs from the trend component. Here, we first evaluate
the scales represented by the mean period in the speciated PM$_{2.5}$ time series. Note that the mean period is only one
indication of the model evaluation against observations, and it does not indicate any information on the magnitude or
the phase of the time series, which will be further discussed in Sections 4.3 to 4.4.
Fig. 3a presents the characteristic scales of IMFs in observed and simulated total and speciated PM$_{2.5}$ of QURE. The
CMAQ model compares well with the observations for IMFs 1 through 6 with cycles of 9, 19, 37, 78, 158 and 347
days (average of all observed and simulated total and speciated PM$_{2.5}$). Among all these IMFs, IMF6, which represents
the annual cycles, shows the least variations in the characteristic scale (Fig. 3a) and highest peak energy from the
power spectrum such as Fig. 2b for total PM$_{2.5}$, except for observed EC and OC where the power of half-year cycles
is more dominant (Fig. S4). These two features demonstrate a clear seasonality in both observed and simulated total
and speciated PM$_{2.5}$, which would otherwise be concealed by practices such as monthly averaging. This can be further
confirmed by the statistically significant annual cycles (except for observed EC and OC) (Fig. S5) based on a Monte
Carlo verified relationship between the energy density and mean period of IMFs (Wu and Huang, 2004; Wu et al.,



2007). To explore the inter-annual cycles in more detail, mean periods of IMFs with scales longer than a year are
being displayed in the top left panel of Fig. 3a. Some variability exists between the observation and model simulation
to the extent that not all IMFs from observation are being simulated and vice versa. The estimated mean periods of
the inter-annual cycles and the differences in the presence of slow varying cycles with the long characteristic scales
are likely to be influenced by their proximity to the data time span of 6 years (4 years for Cl). This implies that the
model evaluation shouldn't go beyond 3 years (2 years for Cl) given the current data coverage. CMAQ captured the
3-year cycles in EC and total $PM_{2.5}$ and 2-year cycles in OC and Cl, despite an overestimation in the scales of 2-year
cycles in observed $SO_4$ and $NO_3$.

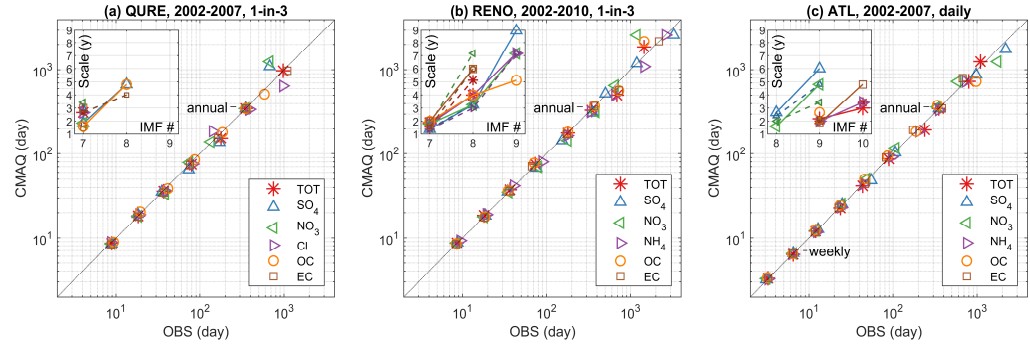


**Fig. 3. The characteristic scales resolved in the IMFs of observed and simulated total and speciated $PM_{2.5}$ for**
**(a) QURE, (b) RENO and (c) ATL. IMF1 to the last pair of IMFs with increasing characteristic periods are**
**shown from bottom left to top right. Top left panel in each subplot shows characteristic scales in the unit of**
**years (y-axis) of all IMFs with inter-annual cycles (the x-axis represents the IMF number). In the subplots,**
**species decomposed from observations are connected by solid lines, while species decomposed from simulations**
**are represented by smaller markers in darker shades connected by dashed lines.**

Similar features in observed and simulated total and speciated $PM_{2.5}$ concentrations at RENO are presented in Fig. 3b.
Likewise, the highest peaks in the power spectrum also sit in the annual cycles of IMF6 except for the observed OC
and total $PM_{2.5}$ which have higher peak power at half-year cycles. All annual IMFs are statistically significant except
for simulated $NH_4$ (Fig. S5). The small variation in the estimated characteristic period of IMF6 is because this
monitoring site is located in a wildfire prone region on the border of Nevada and California. Clear evidence can be
seen from Fig. 4a that an extra annual cycle in the IMF6 of observations in the summer of 2008 is depicted, which is
very possibly driven by the 2008 California Wildfires spanning from May until November. Unlike the diversified
scales in IMF7 at QURE, IMF7 at RENO features universal 2-year cycles of all species as well as total $PM_{2.5}$ and all
of them are well replicated by the model. However, variations in time scales are present in IMF8 possibly because of
the limited data coverage. Thus, only species with time scales less than 4 years in both observations and model
simulations are evaluated. It is evident that CMAQ has reproduced the 3-year cycles in $SO_4$ and $NH_4$.



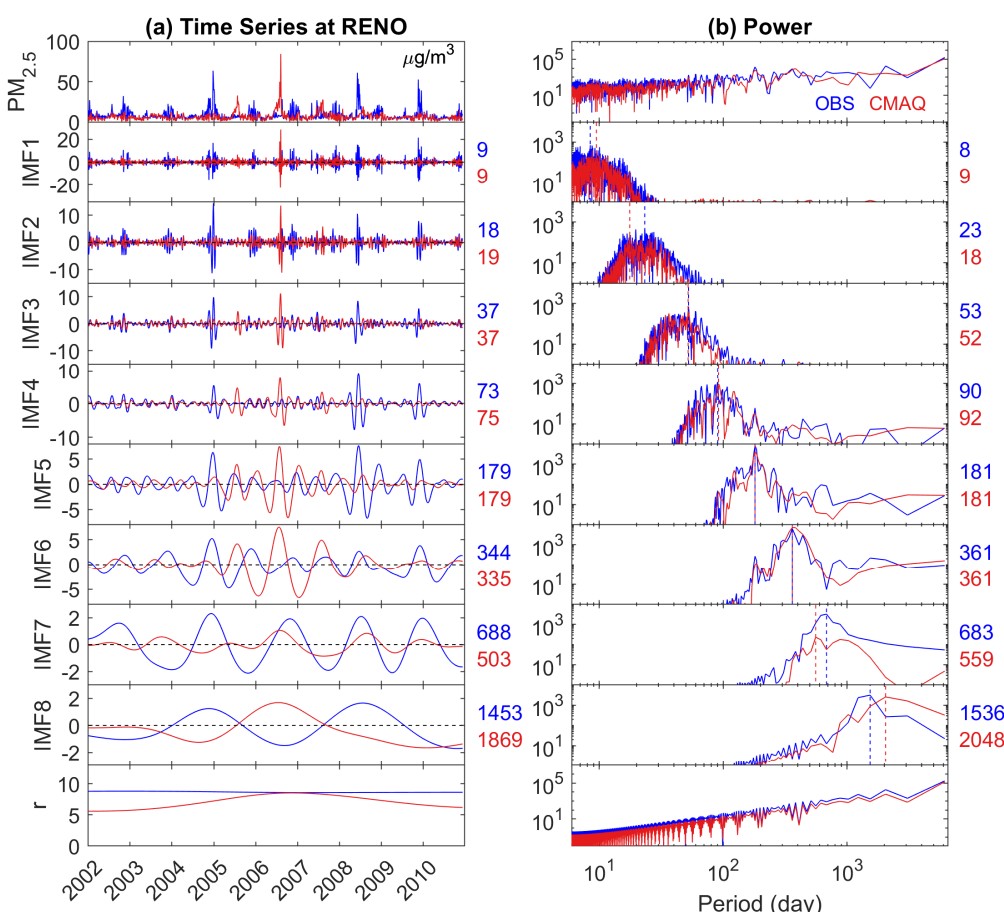


**Fig. 4. Same as Fig. 2 but for the RENO site with 8 IMFs.**

ATL is the only speciated site with daily data coverage. Observed and simulated total and speciated PM$_{2.5}$
concentrations at the ATL site are decomposed into 9 or 10 IMFs (Fig. 3c). Because of the change in data frequency,
high frequency scales such as weekly cycles can be evaluated and the significance tested (Fig. S5) annual cycles with
the highest peak power is represented by IMF8(IMF7 for SO$_4$ and NO$_3$). Annual cycles of SO$_4$ and NO$_3$ appeared in
the earlier stage of decomposition in IMF7 because of their relatively weak half-year cycles, which largely led to the
mixed signal of half year and annual cycles in IMF7 in total PM2.5 as in Fig. 5b. This is more visible in the observed
IMF7 where the energy of the one-year period surpasses that of the half year. Yet, clues can be seen from Fig. 5 that
the amplitude and the energy of annual cycles leaked into IMF7 is very limited compared to that remaining in IMF8,
indicating that it is still safe to conduct model evaluation on the seasonality using IMF8 with an underestimation in





the amplitude of observation. On the other hand, inferences should be made with caution for IMF7 because of the
mixed modes. Scales up to 3 years are relatively well reproduced by the model.

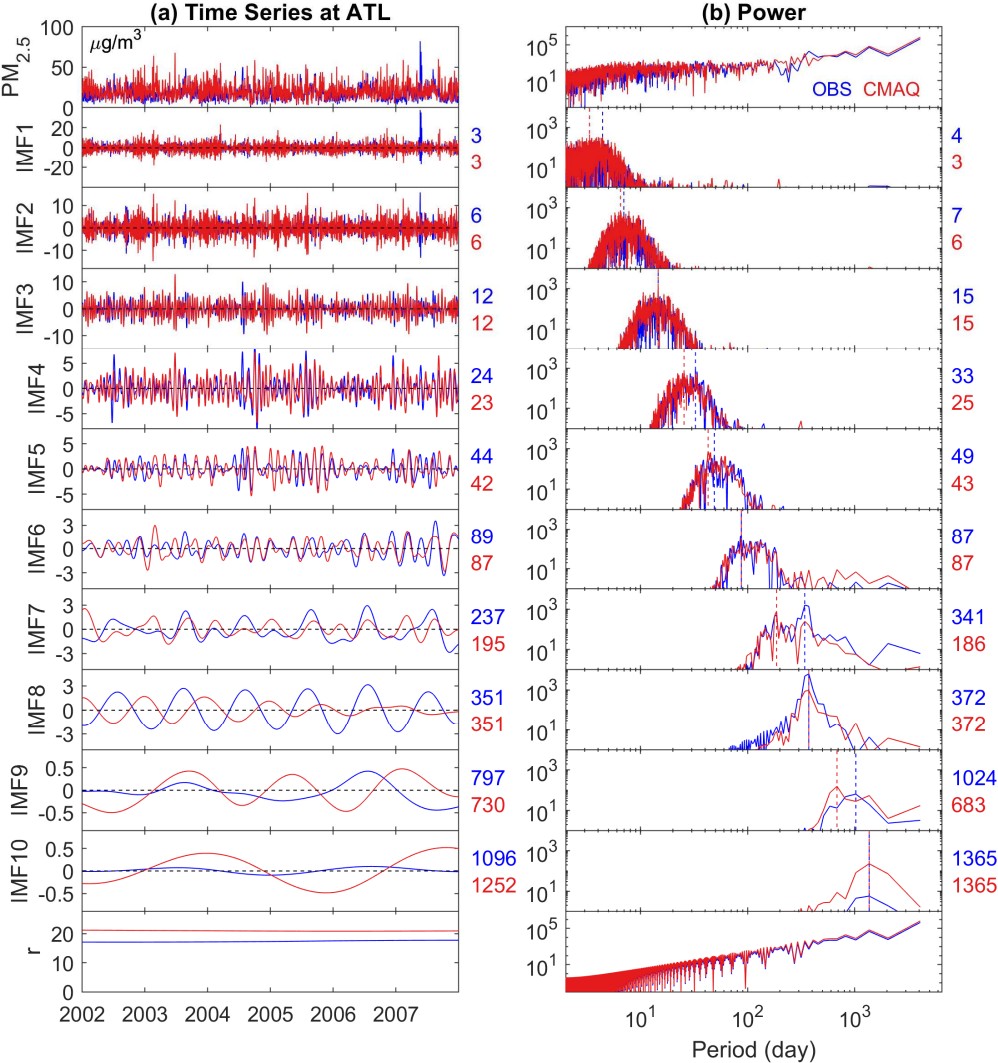


**Fig. 5. Same as Fig. 2 but for the ATL site with 10 IMFs.**

## 4.2 Long-term trend

The EMD-decomposed long-term trend components for the observed and simulated total and speciated $PM_{2.5}$
concentrations are presented in Fig. 6. To better visualize the non-linearity of the trend component, the rates of change
(temporal derivative of a trend component, which is the change in the consecutive concentration divided by the



sampling rate of 1 or 3 days and converted to the unit of µg/m³/year by multiplying 365 day/year) are added with a
separate y-axis on the right side in each panel (gray colored scale). It is evident that PM$_{2.5}$ is changing at a varying
rate, forming either a monotonic trend component or a trend component with one extremum, which cannot be fully
represented by a single constant number using a traditional linear regression approach. Given that there are chemical
species other than the ones studied in the total PM$_{2.5}$, not all performance issues can be fully explained by the five
available species.

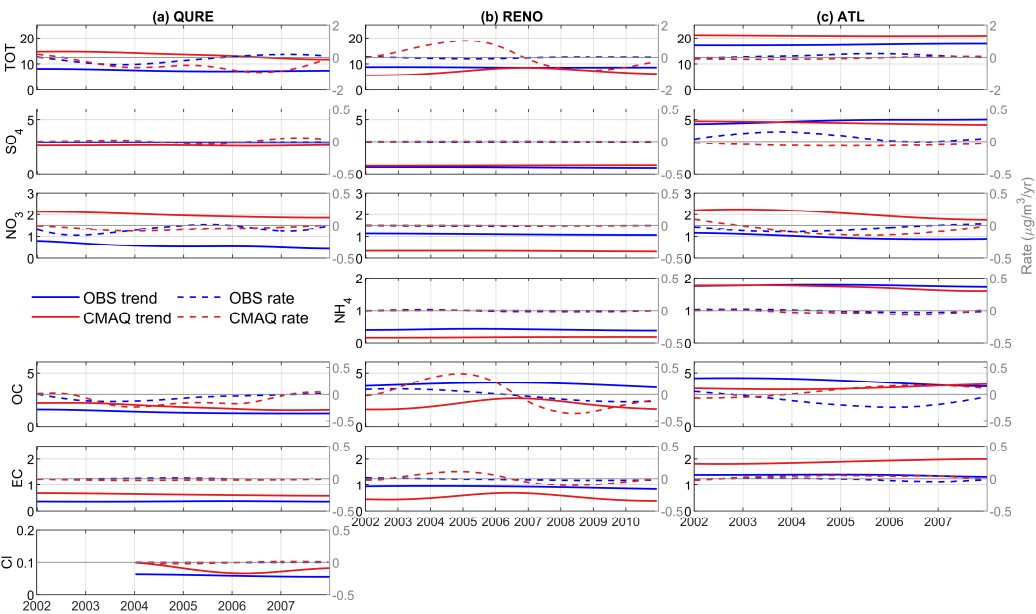


**Fig. 6. Trend components of observed and simulated total and speciated PM$_{2.5}$ for (a) QURE, (b) RENO and (c) ATL in µg/m³ with dashed lines representing the rate of the change (temporal derivative of the trend component converted to µg/m³/year) against the right-side y axis, with a reference line of no change in dark gray line in the center.**

At the QURE site, CMAQ captures the general decreasing trend in observed total PM$_{2.5}$ which can mainly be traced
back to NO$_3$ and OC, while both observed and simulated trend components in SO$_4$ and EC are relatively constant (Fig.
6a). Moreover, the periods with highest decreasing rate in observed total PM$_{2.5}$ during 2003-2004 with a decreasing
rate of -0.44 µg/m³/year is also well replicated by the model. Nevertheless, the slightly increasing PM$_{2.5}$ level in the
later years is simulated to be decreasing at a much higher rate, which is partly due to the overestimated decreasing rate
in OC and species other than the five studied ones. The trend component of simulated Cl shows a cyclic-like feature
because of proximity between the existence of a cycle of 4-5 years (by decomposing the simulation during the 6-year
study period) and 4-year period limited by the available quality assured observations. The rate of change in the
simulated trend component by decomposing the simulation during the 6-year study period would mimic that from the



304 4-year observation, both with a negligible negative value throughout 2004-2007. However, the magnitude of the trend

305 component is almost doubled (1.8 times compared with observation) in the model with contribution from all species

306 except for $SO_4$. A quantitative summary of the magnitude of the trend component can be found in Table 1.

307 **Table 1.** The ratio of mean magnitude of the trend component $r_{trend}$ (CMAQ/observation). Boldface values indicate
308 a relatively good estimate of the magnitude (0.7 - 1.3). "-" indicates the data is not available (same applies for Tables
309 2 and 3).

|        | TOT | $SO_4$ | $NO_3$ | $NH_4$ | OC  | EC  | Cl  |
|--------|-----|--------|--------|--------|-----|-----|-----|
| QURE   | 1.8 | **0.9** | 3.5   | -      | 1.4 | 1.7 | **1.3** |
| RENO   | **0.8** | **1.3** | 0.3 | 0.4    | 0.5 | 0.6 | -   |
| ATL    | **1.2** | **1.0** | 2.1 | **1.0** | **0.9** | 1.4 | -   |

310

311 RENO is located close to the border with California and is affected by large wildfire breakouts in the western U.S. as

312 can been seen in the spikes of the observed total $PM_{2.5}$ (Fig. 4a). The model simulates large increasing rate up to 1.03

313 $\mu g/m^3$/year and decreasing rate up to -0.80 $\mu g/m^3$/year before and after the 2006-2007 winter season and fails to

314 reproduce the relative stable condition seen in the observations with only -0.09 $\mu g/m^3$/year decreasing in 2004-2005

315 and 0.04 $\mu g/m^3$/year  increasing in 2008-2009 (Fig. 6b). Similar feature is found for combustion related OC and EC

316 species. The observed slightly decreasing trends in $SO_4$ and $NH_4$ during 2005-2009 are not being captured in the model

317 simulations. The magnitude of the trend component is slightly underestimated with $r_{trend}$ of 0.8 with contribution

318 from all species except for $SO_4$ as well (Table 1).

319 During the period of 2002-2007, observations at ATL reveal a slightly increasing $PM_{2.5}$ trend that cannot be explained

320 by the five listed PM2.5 components trend (Fig. 6c), possibly indicating a contribution of the remaining species such

321 as the non-carbonaceous portion of organic matter. Non-carbonaceous organic matter can account for more than half

322 of total organic matter, which, in turn, can account for a large portion of the total $PM_{2.5}$ mass (Edgerton et al., 2005).

323 In contrast, the model shows a slight decreasing trend with a peak decreasing rate in 2003 and misses the peak

324 increasing rate of 0.23 $\mu g/m^3$/year in the winter season of 2005. Similarly, reversed trends are also simulated for $SO_4$,

325 OC and EC, while the change rate in $NO_3$ is well captured. Unlike the previous sites, magnitude of trend components

326 in total and speciated $PM_{2.5}$ are well simulated except for EC (1.4 times the observation) and $NO_3$ (2.1 times).

327 To sum up, the long-term trend at QURE is well simulated by the model. The occurrence of large wildfires lasting for

328 several months have significantly impacted the long-term trend component at RENO and the model failed to capture

329 those combustion-related species and total $PM_{2.5}$ primarily due to limitations in the historical data used to specify day-

330 specific wildfire emissions (Xing et al., 2013). Slightly increasing levels of $PM_{2.5}$ and its species observed at ATL are

331 simulated to be slightly decreasing, except for $NO_3$ which is well simulated. The magnitude of the long-term trend

332 components of total $PM_{2.5}$ and $SO_4$ are well represented by CMAQ (Table 1). The model performs differently across

333 the sites in terms of the magnitudes of the trend component in $NO_3$, $NH_4$, Cl, OC and EC. Species other than those in




the available dataset may also play a considerable role in driving the agreements or disagreements between model
simulations and observations of total PM$_{2.5}$.

## 4.3 Seasonality

The EMD-assisted seasonality evaluations utilize the decomposed IMF with characteristic period of one year to
evaluate the amplitude and phase of the model simulation, both of which are time- dependent. We first demonstrate
the evaluation for total PM$_{2.5}$ at QURE (Fig. 7a). The top panel shows the annual cycle components and the bottom
panel shows its TDIC pyramid. The decreasing amplitude of the annual cycles throughout 2002-2007 is almost
perfectly represented with an overall ratio $r_{annual}$ being 1.0 (Table 2). Each pixel in the TDIC pyramid is the
correlation (color-coded) calculated during a period of time $I(t)$ with width of $t_w$ days (y-axis) centered at a specific
day (x-axis) as introduced in Section 3.2. The annual cycle mean periods are identical between CMAQ and
observations (350 days, Fig. 2a IMF6), but there is a phase shift for all years with the entire TDIC pyramid being close
to -1. By shifting the CMAQ annual cycles backward 159 days (almost half year), the overall correlation of the annual
component can reach up to a peak of 0.9 (Table 3).

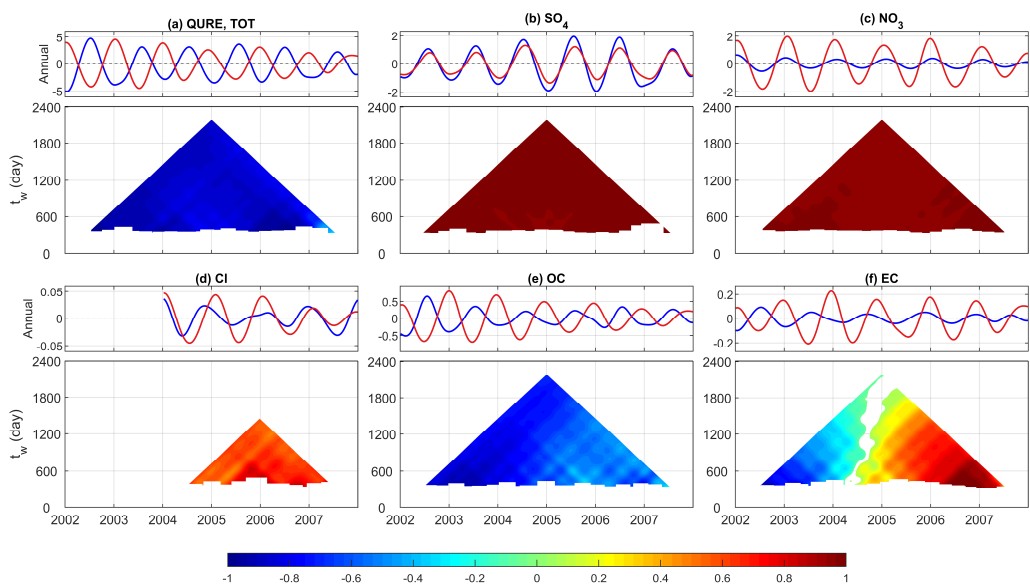

**Fig. 7. Decomposed annual cycles (IMF6) from observed (blue) and simulated (red) concentrations (µg/m³) of (a) total PM$_{2.5}$, (b) SO$_4$, (c) NO$_3$, (d) Cl, (e) OC and (f) EC and their corresponding TDIC at Quabbin Summit, MA. The window size $t_w$ indicates the width of the window used to calculate a specific correlation centered at the day represented in x-axis.**

What are the driving factors for the above phase shift in modeled total PM$_{2.5}$ at Quabbin Summit, MA? The illustrations
in Fig. 7a for total PM$_{2.5}$ alone cannot provide useful information that will allow the modeler to improve the model's
performance. This is accomplished by applying the EMD method to the PM$_{2.5}$ speciated components (Fig. 7b-f). Traces



of the semi-annual phase shift (-159 days) of annual cycles or large overestimation in the winter and underestimation
in the summer is because of the largely overestimated amplitude of $NO_3$ (4.3 times that of observation) which peaks
in the winter and the almost semi-annual shifted OC (-147 days), as well as contributions from EC and Cl. $NO_3$ has a
mean amplitude reaching almost half of that of the total $PM_{2.5}$. OC directly drives both the observed and simulated
annual components to be negatively correlated. EC follows the feature of OC in the first four years or so and the
feature of $NO_3$ in 2006 and 2007 and contributes to the half year shifted total $PM_{2.5}$. The magnitude of winter-peaking
Cl cycles are overestimated with a phase shift of one month. However, the contribution of Cl is very limited because
of the tiny amplitude in both observed and simulated annual cycles. In addition, annual cycles in $SO_4$ are well
reproduced for the entire time span with an amplitude ratio of 0.7. A quantitative summary of the evaluation of the
annual cycles at this site can be found in Tables 2 and 3.

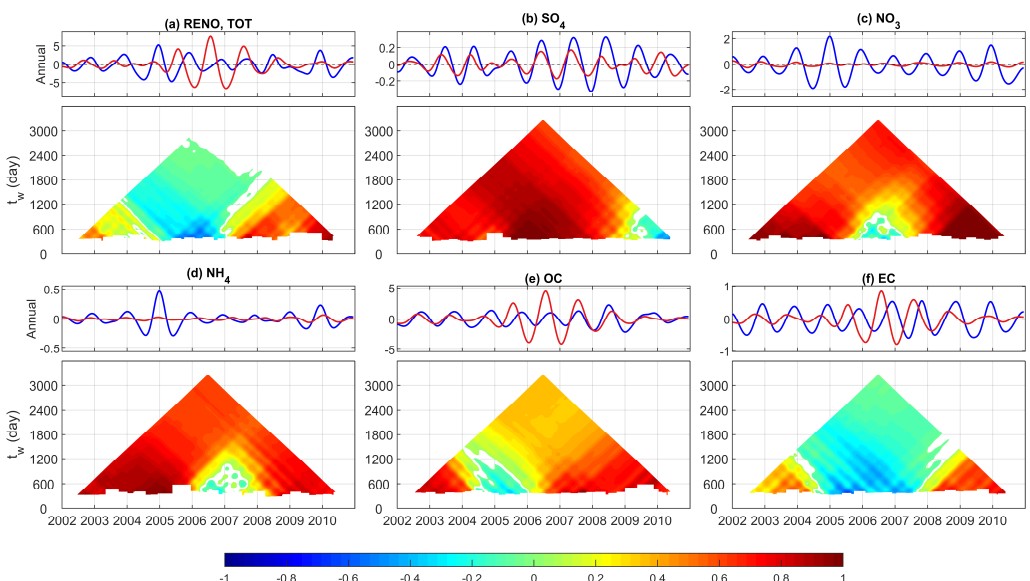


**Fig. 8. Same as in Fig. 7 for Reno, NV, except that (d) represents $NH_4$ rather than Cl.**
Both observed and simulated annual cycles at the RENO site are largely contaminated by the extreme events lasting
for several months that are not properly simulated, indicating the need for more appropriate emissions allocation.
Overall, annual variations for total and speciated $PM_{2.5}$ are largely underestimated except for the total $PM_{2.5}$ and
combustion-driven EC and OC from 2005 to 2007 (Fig. 8). The modeled phase of $SO_4$, $NO_3$, $NH_4$ and OC agrees with
that of observation with exception for a length of about two years in each that missed the phasing: 2009-2010 for $SO_4$,
summer 2005-summer 2007 for $NO_3$, 2006-2007 for $NH_4$ and 2004-2005 for OC. It is also notable that the TDIC
pyramid of EC mimics that of total $PM_{2.5}$, implying the existence of errors in modeled EC in processes such as
emissions, transport, and deposition that affected the model performance for total $PM_{2.5}$. In comparison, $SO_4$ and OC
are relatively well simulated with a mean amplitude ratio of 0.5 and 1.5 and a phase shift of 36 and 33 days,
respectively.





Observed annual cycles of total PM$_{2.5}$ at the ATL site features a slightly increasing amplitude of annual variations
from 2002 to 2006 which then decreased to the original state in 2007 (Fig. 9a). Conversely, model-simulated annual
cycles became weaker throughout the period, with an overall $r_{annual}$ of 0.5. As at the QURE site, the simulated annual
components at the ATL site also show a shift of several months (-132 days). Specifically, traces of these phase shifts
or large overestimation in the winter and underestimation in the summer can be seen from the more than doubled
amplitude of NO$_3$ which peaks in winter and underestimated SO$_4$ and NH$_4$ in the warm seasons as well as the -54 days
shifted EC. The anti-correlated remaining species other than those in the available dataset clearly played a role in
driving the discrepancies seen in the total PM$_{2.5}$ annual cycles (Fig. 10). Specifically, the anti-correlation likely points
to an inaccurate representation of the seasonal variation of the non-carbonaceous portion of organic matter due to an
improper representation of organic aerosols in the model version analyzed here; this problem has since been corrected
in more recent releases of the CMAQ model. The underestimated annual variations in the remaining components
closely resemble that of the annual variation in total PM$_{2.5}$. The phase of simulated SO$_4$, NO$_3$, NH$_4$, and OC species is
in good agreement with those in observations and the amplitude of simulated annual cycles in SO$_4$, OC and EC agree
well with that in the observations (Tables 2 and 3).
In sum, annual cycles of PM$_{2.5}$ are also time-dependent and the phase in the annual cycles for total PM$_{2.5}$, OC and EC
reveal a general shift of up to half a year (Table 3); this indicates a potential problem in the allocation of emissions
during this study period and/or the treatment of organic aerosols in this version of the model. CMAQ generally
simulated the phase in SO$_4$, NO$_3$, Cl and NH$_4$ quite well but did not always capture the magnitude of their variations
(Table 2).

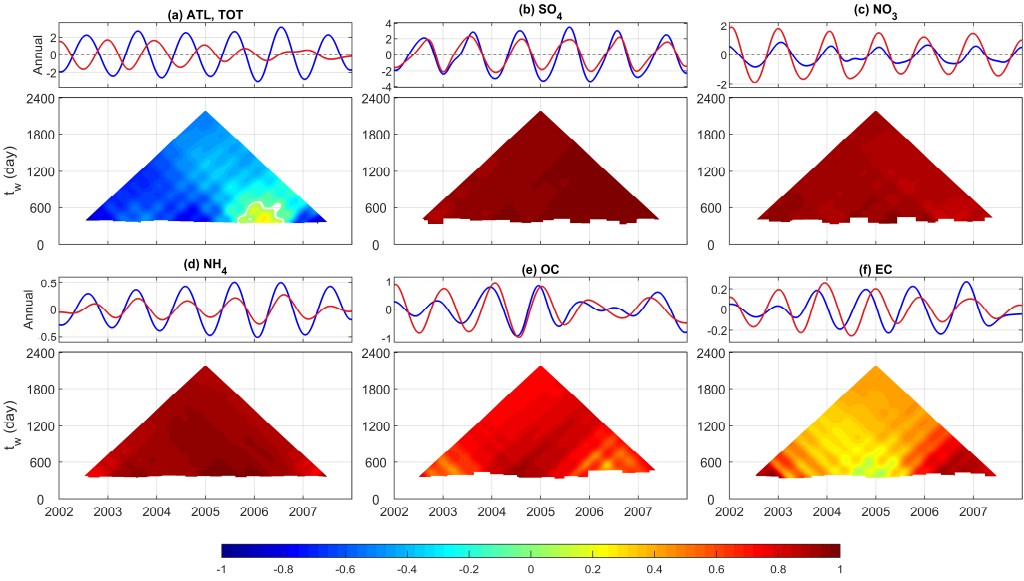






**Fig. 9. Same as in Fig. 7 for Atlanta, GA, except that the annual component is resolved in IMF8 (IMF7 for SO$_4$**
**and NO$_3$) because of the difference in sampling rate and characteristic embedded in the time series at ATL and**
**(d) represents NH$_4$ rather than Cl.**

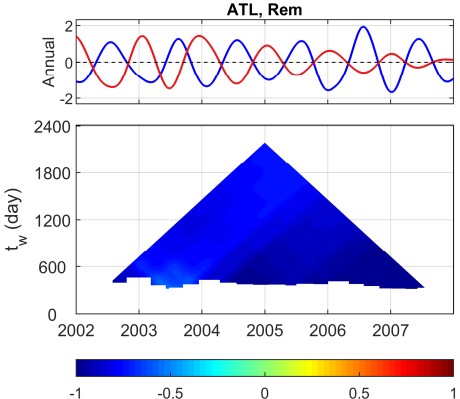


**Fig. 10. Decomposed annual cycles in Atlanta, GA for the remaining components presented in total PM$_{2.5}$ other**
**than the five species in Fig.9.**
**Table 2.** The ratio of mean amplitude of the annual component $r_{annual}$ (CMAQ/observation). Boldface values indicate
a magnitude with a ratio close to 1 (0.7 -1.3).

|        | TOT     | SO$_4$   | NO$_3$ | NH$_4$ | OC      | EC      | Cl  |
|--------|---------|----------|--------|--------|---------|---------|-----|
| QURE   | **1.0** | **0.7**  | 4.3    | -      | 1.6     | 3.1     | 1.6 |
| RENO   | **1.2** | 0.5      | 0.1    | 0.2    | 1.5     | **0.9** | -   |
| ATL    | 0.5     | **0.7**  | 2.4    | 0.4    | **1.2** | **1.0** | -   |


**Table 3.** Phase shift ($n$) of CMAQ simulated annual cycle components in days. The background color indicates the
maximum correlation ($R_{max}$) that can be reached by shifting the CMAQ time series $n$ days with respect to
observations: white = [0.8, 1], light grey = [0.6, 0.8), grey = [0.4, 0.6), dark grey = (0.2, 0.4). The bold shows number
of shifts less than a month while the italic shows shifts longer than three months.

|        | TOT      | SO$_4$  | NO$_3$ | NH$_4$  | OC       | EC       | Cl      |
|--------|----------|---------|--------|---------|----------|----------|---------|
| QURE   | *-159*   | **-6**  | **3**  | -       | *-147*   | *-105*   | **-30** |
| RENO   | 78       | 36      | **12** | **-21** | 33       | *96*     | -       |
| ATL    | *-132*   | **0**   | **8**  | **-17** | **-24**  | -54      | -       |




### 4.4 Sub-seasonal and inter-annual variability


In this section, model performance at multiple sub-seasonal and inter-annual scales with cycles less than 3 years,
presented in the total and speciated $PM_{2.5}$, is evaluated following an approach similar to that for the annual cycles in
Section 4.3 (Fig. 11). First, IMFs from observations and model simulations are paired based on their characteristic
periods following the discussion in Section 4.1. Then, the magnitude of specific scales is evaluated using $r_{IMFn}$
following Equation 6 of the $r_{annual}$ for annual cycles. The phase shifts of the time series are assessed by the proportion
of shifted days relative to the mean characteristic scales of the corresponding observed and simulated IMFs ($^{n}/_{t_{m}}$).
For example, a phase shift of 0.1 cycles in the 2-year cycles is approximately 73 days while it would be 18 days for
the half-year cycles.
The performance of the simulated amplitude of the sub-seasonal and inter-annual cycles is relatively stable from a few
days to semi-annual scales and $r_{IMFn}$ is close to 1 in most cases (Fig. 11a-c). CMAQ captures the features seen in the
observations at QURE, except for the large overestimation of $NO_3$ ($r_{IMFn}$ ranges from 2.6 to 3.7 at the sub-seasonal
scale and reaches up to 13.8 for the 3-year cycles). Similar overestimation of $NO_3$ is also found at ATL ($r_{IMFn}$ ranges
from 2.0 to 3.4, except for the 2-year cycles). In contrast, $NO_3$ at RENO is strongly underestimated with $r_{IMFn}$ ranging
from 0.1 to 0.3 and reaching its minimum at the 2-year cycles. Likewise, all time scales of $NH_4$ at RENO are also
being underestimated with $r_{IMFn}$ decreasing from 0.4 to only 0.1 at the 3-year cycles. The coexistence of
underestimation of $NO_3$ and $NH_4$ variability, as well as their trend component, likely points to the insufficient grid
resolution in representing ammonium nitrate episodes associated with stagnant meteorology in the mountainous
regions as illustrated by Kelly et al. (2019). To sum up, model has simulated the magnitude of features across all scales
in most of the studied cases. However, fluctuations in $NO_3$ are constantly being largely over- or under-estimated and
improvements to the model are required to better replicate its variability (Fig. 11a-c).
A high $R_{max}$ of corresponding IMFs can only be achieved when the characteristic scales of those from observations
and model simulations are close, there is minimal mode mixing, and negligible irregular change of amplitude exists
during the study period. Thus, $R_{max}$ tends to be small for all oscillations at RENO because of the irregular impact
from events such as wildfires. Thus, the interpretation of phase shift is focused on the components and time scales
having correlations above 0.4 only.
Results show that the sub-seasonal cycles at QURE all have a negligible phase shift of less than 0.1 cycles (Fig. 11d).
The semi-annual cycles at RENO have around 0.2 cycle phase shifts in total $PM_{2.5}$ (-0.2), $NH_4$(0.2), OC (-0.2), and
EC (-0.2) while negligible phase shifts of less than 0.1 cycles are simulated in $SO_4$ ranging from 9 days to semi-annual
in scale. As at QURE, multiple sub-seasonal cycles at ATL all have a negligible phase shift of less than 0.1 cycles,
with the exception of semi-annual OC which has a phase shift of nearly -0.4 cycles with a marginal correlation of
around 0.4. Unlike the relatively stable $R_{max}$ throughout the time scales within each of the species for QURE and
RENO, the $R_{max}$ at ATL tends to be much higher (roughly 0.6-0.8) in the scales of 6 to 25 days, except for $NO_3$,
indicating the model's success in simulating those weather-induced air quality fluctuations at this site as reflected by
their negligible phase shifts.



However, the physical meaning of each sub-seasonal IMF is not yet fully understood and requires further study.
Synoptic scale IMFs (IMFs with scale less than/around a month) usually have large variance and are not statistically
significant different from white noise except for observed $SO_4$ and $NH_4$ (Fig. S5). Yet, observed and simulated total
and some speciated $PM_{2.5}$ at QURE and ATL (except IMF1) can achieve moderate to high $R_{max}$ at these time scales
(Fig. 11 g-i), indicating a potential physical explanation of those time scales using meteorological variables. IMFs
with scales longer than a month but less than half year possess much less variance and are usually not statistically
significant different from noise. Exceptions are also found at the Atlanta site where observed IMFs are mostly
significant different from noise. Whereas semi-annual cycles are mostly statistically significant (note that semi-annual
$SO_4$ and $NO_3$ at ATL are too weak to be decomposed into a separate IMF). In a previous study, He et al. (2014) found
semi-annual oscillations in the corrected AErosol RObotic NETwork (AERONET) Aerosol Optical Depth (AOD) and
$PM_{10}$ mass concentrations are primarily caused by the change of wind directions in Hong Kong.

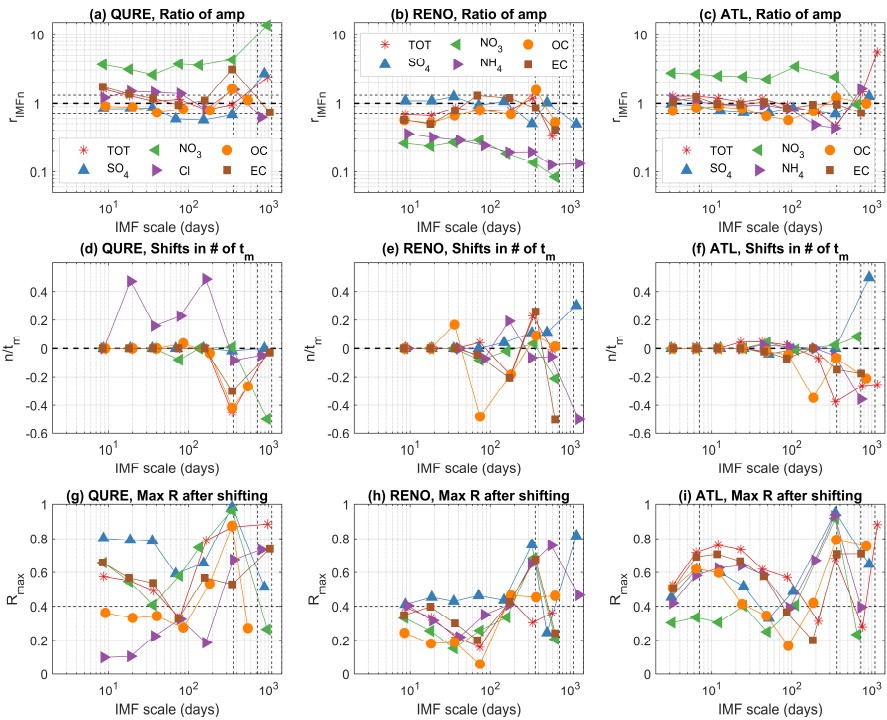


**Fig. 11. Model performance at all temporal scales for sites QURE, RENO and ATL. (a-c) ratio of mean**
**amplitude of corresponding IMFs with similar characteristic mean periods (ideal ratio=1.0); (d-f) the phase**
**shift $n$ in the number of mean periods (average mean period of corresponding IMFs decomposed from**
**observation and model simulation); (g-i) maximum correlation $R_{max}$ can be achieved by shifting the modeled**
**time series. The average mean period of corresponding IMFs decomposed from observations and CMAQ of**
**total and speciated $PM_{2.5}$ are represented on the x-axis; all metrics on the y-axis are unitless. Horizontal**



**reference lines are drawn at 0.7 and 1.3 in (a-c). Weekly, annual and inter-annual (2- to 3-year) scales are**
**marked with vertical dashed lines.**
The evaluation and interpretation of inter-annual cycles are constrained by the limited available speciated observations
for a period of 6 to 9 years (4 years for Cl at QURE). Thus, only 2- to 3-year cycles are presented (Fig. 11) and
evaluated. Among the 2- to 3-year inter-annual cycles at QURE, there is minimal phase shift for total $PM_{2.5}$, $SO_4$, Cl,
and EC with moderate to high $R_{max}$. At RENO, the model presents negligible shifts in 2-year cycles of OC and $NH_4$
while phase shifts of 0.3 and -0.5 cycles are simulated in the 3-year cycles for $SO_4$ and $NH_4$. At ATL, the phase shift
of -0.2 to -0.4 cycles are simulated for $PM_{2.5}$, $NH_4$, OC, and EC with periods of 2- to 3-year cycles; while 2- to 3-year
$SO_4$ cycles have a half-year cycle shift.

## 474 5 Conclusions

The main advantage for using EMD to evaluate $PM_{2.5}$ and its speciated components is that it decomposes nonlinear
and nonstationary signals into multiple modes and a residual trend component. It does not require any preselection of
the temporal scales and assumptions of linearity and stationarity for the data, thereby providing insights into time
series of $PM_{2.5}$ concentrations and its components. Using improved CEEMDAN, we are able to assess how well
regional-scale air quality models like CMAQ can simulate the intrinsic time-dependent long-term trend and cyclic
variations in daily average $PM_{2.5}$ and its species. This type of coordinated decomposition and evaluation of total and
speciated $PM_{2.5}$ provides a unique opportunity for modelers to assess influences of each $PM_{2.5}$ species to the total
$PM_{2.5}$ concentration in terms of time shifts for various temporal cycles and the magnitude of each component including
the trend.
A demonstration of how improved CEEMDAN could be applied to time series data at three sites over CONUS that
provide speciated PM2.5 data reveals the presence of the annual cycles in $PM_{2.5}$ concentrations and time-dependent
features in all decomposed components. At these three sites, the model generally is more capable of simulating the
change rate in the trend component than the absolute magnitude of the long-term trend component. However, the
magnitude of $SO_4$ trend components is well represented across all three sites. Also, the model reproduced the amplitude
of the annual cycles for total $PM_{2.5}$, $SO_4$ and OC. The phase difference in the annual cycles for total $PM_{2.5}$, OC and
EC reveal a shift of up to half-year, indicating the need for proper allocation of emissions and an updated treatment
of organic aerosols compared to the earlier model version used in this set of model simulations. The consistent large
under/over-prediction of $NO_3$ variability at all temporal scales and magnitude in the trend component, as well as the
abnormally low correlations of synoptic scale $NO_3$ at ATL, calls for better representation of nitrate partitioning and
chemistry. Wildfires have the potential to elevate $PM_{2.5}$ for months and can alter its variability at scales from few days
to the entire year. Thus, more accurate fire emission data should be incorporated to improve model simulation,
especially in those fire-prone regions.
**Data availability**. Paired observations and CMAQ model data used in the analysis will be made available at
https://edg.epa.gov/metadata/catalog/main/home.page. Raw CMAQ model outputs are available on request from the
U.S EPA authors.



**Author contribution.** "HL and MA designed the methodology; RM, CH and SR contributed in the assessment of the outcomes and were consulted on necessary revisions. Model simulations were performed by the US EPA. HL prepared the manuscript with contributions from all co-authors."

**Acknowledgements**

The views expressed in this paper are those of the authors and do not necessarily represent the view or policies of the U.S. Environmental Protection Agency. Two of the authors (MA and HL) acknowledge that part of this work was supported by the Electric Power Research Institute (EPRI) Contract #00-10005071, 2015–2017.

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
