# Peer review of "Evaluating Trends and Seasonality in Modeled PM2.5 Concentrations Using Empirical Mode Decomposition"

_Atmospheric Chemistry and Physics, 2019_

## Referee Comment (RC1) · Anonymous Referee #3 · 7 Jun 2020

This manuscript presented an evaluation of the WRF-CAMQ model simulated temporal trends through a detailed comparison with observation using improved CEEMDAN method. The comparison was based on measurements of PM2.5 and its key components, i.e., sulphate, nitrate, ammonium, chloride, organic carbon, and elemental carbon, made at three ground monitoring stations in US from t 2002 to 2008. It is clearly demonstrated that the improved CEEMDAN approach can decompose the observed and simulated temporal trends, which allows to extract more information from the comparisons of individual temporal modes. For example, the authors concluded that the model can better simulate the rate of change of the multi-year trend than the absolute magnitude. At the same time, model can generally reproduce the amplitudes

of the sub-seasonal and annual variations for PM2.5, sulphate, and OC. This study revealed that it appears there is a temporal phased shift between the observed and model simulated PM2.5, OC, and EC as large as a half year. It is further suggested that this phase shift indicted "a need for proper temporal allocation of emissions". In general, the manuscript is well organized. This reviewer believes that this is an important work which can potentially help identifying model deficiencies. However, there several concerns needed to be addressed: 1) The authors correctly stated that EMD is a widely used methodology in various field. At the same time, this reviewer would like to suggest that the authors should consider adding some brief high-level descriptions of the method. This will improve the manuscript's readability, especially for those who are not familiar with EMD methods. It is also important to clearly state the criteria how the modes are determined and separated. The statement in line 134-135, "to achieve best mode separation", leaves much room for interpretation. The discussion on determination of tp and tm is interesting and thorough. It does, however, leave an impression that the evaluation of tp and tm is somewhat uncertain and is not completely deterministic. This reviewer would like to suggest adding additional text to discuss if the determination of tp and tm is sufficiently accurate or useful for model assessment to identify issues in the processes at the similar time scale as decomposed tp and/or tm. This will strengthen the manuscript to demonstrate the usefulness of the improved CEEMDAN approach in model assessments. 2) Section 2 (starting from line 74) provided a good discussion on how the observation data sets are selected. It is equally important to discuss the temporal resolution of model in terms of the driving factors, e.g., emissions. This will give readers a sense if one should expect if the model should reproduce observations at certain temporal scale. For example, if the emissions are given in yearly average, one would consider the impact of the lack emission temporal variability on the comparison of seasonal and/or sub-seasonal trends. 3) This reviewer believes that the concluding remark of "indicating the need for proper allocation of emissions" is an important conclusion. However, it was not adequately justified. There are many controlling factors and processes. The authors should have provided more discussions to

illustrate how they narrowed to emissions as the likely factor. It should also be pointed out that SOA is typically a large component of OC. Changes in emissions to affect OC will likely have implications on O3. 4) The authors presented detailed trend analysis on PM2.5 and its components. It is also scientifically interesting to understand the relative contribution of each component and their contribution to the identified temporal variability, which are useful to gain insights into controlling factors. This reviewer would like to suggest the authors to consider addition of the trend analysis on the relative contribution of sulphate, nitrate, ammonium, organic carbon, and elemental carbon to PM2.5. More specific to the manuscript, it would be much easier to interpret the results shown in Table 1, 2, and 3 if the relative contribution of each component is known. 5) In general, model evaluation is designed to improve model. It is difficult to relate the comparison results presented in this manuscript to specific model deficiencies in description of the chemical/physical processes and/or issues in model data sets, meteorological field and/or emission data. As sulphate, OC, nitrate are controlled by very different chemical processes, this reviewer would like to encourage the authors to further explore the difference in the comparison results for these species, which may reveal additional insights into the process-level model deficiencies.

Specific Comments: 1) Figure 3 is hard to read because of log-log scale. It may be better to change the x-axis to the IMF number and y-axis to the ratio between model and observation characteristic scales. A second y-axis can be added to show the absolute characteristic scales for each IMF. 2) Section 4.2. Figure 6 shows some variation in time-derivatives. At the same, this reviewer would like to argue that about half of cases shown in the figure can be well approximated by linear assumption. The authors should comment on this aspect.

---

## Referee Comment (RC2) · Anonymous Referee #4 · 19 Jun 2020

Comment on "Evaluating Trends and Seasonality in Modeled PM2.5 Concentrations Using Empirical Mode Decomposition"

General comments: This paper introduces a new approach for process-based model evaluation of speciated PM2.5, which allows for the assessment of the performance of regional-scale air quality models like CMAQ on the intrinsic time-dependent long-term trend and cyclic variations in daily average PM2.5 and its species. The authors tested the method with time series data at three sites. The data are generally sound, whereas some results and discussions of the study are still lack of persuasion. One major concern is about how well the current approach's performance is compared with

the previously published methods and some over-interpreted conclusions. The other is that it is not sure that the difference between the model and the new approach evaluation results can be simply explained by the inadequate description of nitrate or organics in the model. As the authors noted, they obtained abnormally low correlations of synoptic scale NO3 at ATL and calls for a better representation of nitrate partitioning and chemistry. What about the results for the other two sites? The authors need to provide more information on such issues to make the conclusion robust.

Specific comments:

1. Introduction: "Evaluation of ten-year averaged monthly mean of PM2.5 simulated with WRF/Chem . . ." how does the model performance of PM2.5 compositions simulation should also be summarized to provide an intact view on the previous results.

2. Line 36: "and other natural species..." what do natural species refer to?

3. Line 47: "monthly or seasonal means" means of speciated PM2.5?

4. Line 48: what do you mean by "ten-year averaged monthly mean"?

5. Line 51: "with a phase shift of few months" please explain phase shift.

6. Line 55-57: ". . .long-term trends or interannual variations driven by climate change, emission control policies or other slow varying processes..." what is the main reason? Are there any previous results?

7. Line 68-74: I do not think this paragraph is necessary for the manuscript.

8. Line 311: "RENO is located close to the border with California and is affected by large wildfire breakouts in the western U.S. . .." Is there any evidence for this demonstration?

9. Line 327-: "To sum up, the long-term trend at QURE is well simulated by the model." This is unlikely consistent with the data presented in Table 1.

10. Lines 333-335: "the available dataset may also play a considerable role in driving the agreements or disagreements between model simulations and observations of total PM2.5" What are the contribution of these species to PM2.5 at the studied sites?

11. Lines 367-368: "Both observed and simulated annual cycles at the RENO site are largely contaminated by the extreme events lasting for several months that are not properly simulated" is it possible to remove the data of extreme events before simulation, in order to eliminate the contamination?

12. Lines 384-387: "Specifically, the anti-correlation likely points to an inaccurate representation of the seasonal variation of the non-carbonaceous portion of organic matter due to an improper representation of organic aerosols in the model version analyzed here; this problem has since been corrected in more recent releases of the CMAQ model." This sentence needs to be rewritten for clearance. And what does the non-carbonaceous portion of organic matter refer to?

Minor:

Line 17: "chloride (Cl) organic"

Line 311: "U.S. as can been seen"

---

## Author Comment (AC1) · 31 Jul 2020

Reply to interactive comments on "Evaluating Trends and Seasonality in Modeled PM2.5 Concentrations Using Empirical Mode Decomposition"

Anonymous Referee #3

This manuscript presented an evaluation of the WRF-CAMQ model simulated temporal trends through a detailed comparison with observation using improved CEEMDAN method. The comparison was based on measurements of PM2.5 and its key components, i.e., sulphate, nitrate, ammonium, chloride, organic carbon, and elemental

carbon, made at three ground monitoring stations in US from t 2002 to 2008. It is clearly demonstrated that the improved CEEMDAN approach can decompose the observed and simulated temporal trends, which allows to extract more information from the comparisons of individual temporal modes. For example, the authors concluded that the model can better simulate the rate of change of the multi-year trend than the absolute magnitude. At the same time, model can generally reproduce the amplitudes of the sub-seasonal and annual variations for PM2.5, sulphate, and OC. This study revealed that it appears there is a temporal phased shift between the observed and model simulated PM2.5, OC, and EC as large as a half year. It is further suggested that this phase shift indicted "a need for proper temporal allocation of emissions". In general, the manuscript is well organized.

We thank the reviewer for the positive assessment of our manuscript and for providing constructive feedback to help improve the quality of the manuscript. We have addressed all questions and suggestions in our response as well as in the text or figures, as necessary. Please see detailed responses below and the marked-up version of the revised manuscript.

This reviewer believes that this is an important work which can potentially help identifying model deficiencies. However, there several concerns needed to be addressed:

1) The authors correctly stated that EMD is a widely used methodology in various field. At the same time, this reviewer would like to suggest that the authors should consider adding some brief high-level descriptions of the method. This will improve the manuscript's readability, especially for those who are not familiar with EMD methods. It is also important to clearly state the criteria how the modes are determined and separated. The statement in line 134-135, "to achieve best mode separation", leaves much room for interpretation. The discussion on determination of tp and tm is interesting and thorough. It does, however, leave an impression that the evaluation of tp and tm is somewhat uncertain and is not completely deterministic. This reviewer would like to suggest adding additional text to discuss if the determination of tp and tm is sufficiently accurate or useful for model assessment to identify issues in the processes at the similar time scale as decomposed tp and/or tm. This will strengthen the manuscript to demonstrate the usefulness of the improved CEEMDAN approach in model assessments.

The decomposition process and parameters controlling the decomposition have been added in Section 3.1 as suggested. The "best mode separation" is also further explained following the reviewer's suggestion.

CEEMDAN is a technique that is particularly suitable to analyze non-linear and non-stationary time series data. The decomposed time series of speciated and total PM2.5 reveal the agreement/disagreement between observations and model simulations at various intrinsic temporal scales without any predetermined assumptions on the data. Both tp and tm represent approximate estimates of the characteristic scale of an IMF, where non-linear and non-stationary processes with close temporal scales could exist. For tp (from the revised text): "The peak estimates can be biased if more than one high-power frequency is located closely within one IMF. Thus, the power spectrum and t_p is only used as a fast screening tool to determine if a desired decomposition is accomplished." For tm: "As the frequency decreases, the mean period estimates become less accurate because of the limited time span compared with the length of the cycle and should be carefully interpreted." We have added the following test in Section 4.1: "Since each IMF represents a non-stationary process, the mean period t_m is only an estimate of its characteristic scale. Evaluation of t_m might not necessarily be able to identify issues with corresponding model simulations, and it does not indicate any information on the magnitude or the phase of the time series, which is more important and will be further discussed in Sections 4.3 to 4.4.".

2) Section 2 (starting from line 74) provided a good discussion on how the observation data sets are selected. It is equally important to discuss the temporal resolution of model in terms of the driving factors, e.g., emissions. This will give readers a sense if one should expect if the model should reproduce observations at certain temporal

scale. For example, if the emissions are given in yearly average, one would consider the impact of the lack emission temporal variability on the comparison of seasonal and/or sub-seasonal trends.

We added the following text in Section 2: "Annual emissions for the CMAQ simulations were estimated using the methodology described in Xing et al. (2013). Briefly, the National Emissions Inventory (NEI) for 1990, 1995, 1996, 1999, 2001, 2002 and 2005 and a number of sector-specific long-term databases containing information about trends in activity data and emission controls were used to create county-level annual emissions for a total of 49 emission sectors. Prior to being used as input to the CMAQ simulations, these annual emissions were then temporally and spatially allocated to provide hourly emissions based on monthly, weekly, and diurnal temporal cross-reference and profile data from the 2005 NEI modeling platform. These profile data vary by emissions source and sometimes by state and county and are generally based on surveys and extrapolation of activity data which can be subject to uncertainty. Exceptions to the use of 2005 NEI platform temporal profile data for temporal allocation were emissions from electric generating units (EGU) which directly used measured hourly emissions after 1995 and wildfire emissions that used climatological monthly, weekly, and diurnal profiles for temporal allocation."

The large discrepancy in the magnitude of some long-term trend component seen in Fig. 6 is likely pointing to the systematic bias in the annual emission estimations as discussed in Xing et al. (2013): "…since this study mainly focused on trends rather than the absolute value in each individual year, some sectors (e.g., industrial processes) and sub-sectors (types of combustion and stoves) may not have been well considered or examined." The intra-annual emission allocation could possibly impact the model performance at the seasonal and sub-seasonal scales. Thiss discussion of the impact of emissions on the long-term trend has been added in Section 4.2.

3) This reviewer believes that the concluding remark of "indicating the need for proper allocation of emissions" is an important conclusion. However, it was not adequately

justified. There are many controlling factors and processes. The authors should have provided more discussions to illustrate how they narrowed to emissions as the likely factor. It should also be pointed out that SOA is typically a large component of OC. Changes in emissions to affect OC will likely have implications on O3.

We would like to clarify that our illustrative application of the new methodology to PM2.5 time series at three specific sites does not allow us to conclude that errors in the temporal allocation of PM emissions are the controlling factors for disagreements between observed and modeled annual cycle. While we believe that they do play a role as discussed below, we also know that the CMAQ version used for these simulations has underestimated the formation of SOA, which would also affect the modeled annual cycle of OC (e.g. Appel et al., 2017; Murphy et al., 2017; Xu et al., 2018). Because of the underestimation of SOA, OC in the simulations analyzed here has an overestimated relative contribution of primary OC which, in turn, makes its temporal variations analyzed by CEEMDAN sensitive to the temporal allocation of primary PM and specifically primary OC emissions. The full statement partially quoted by the reviewer points to both factors "indicating the need for proper allocation of emissions and an updated treatment of organic aerosols compared to the earlier model version used in this set of model simulations". Without running a new set of decadal simulations with a newer version of the model and/or modified temporal allocation of emissions, we are unable to determine the relative importance of these factors at the sites examined. However, if such simulations were to be performed in the future, the CEEMDAN methodology can help demonstrate the benefits of updated emissions allocations and/or the SOA process representation.

4) The authors presented detailed trend analysis on PM2.5 and its components. It is also scientifically interesting to understand the relative contribution of each component and their contribution to the identified temporal variability, which are useful to gain insights into controlling factors. This reviewer would like to suggest the authors to consider addition of the trend analysis on the relative contribution of sulphate, nitrate,

ammonium, organic carbon, and elemental carbon to PM2.5. More specific to the manuscript, it would be much easier to interpret the results shown in Table 1, 2, and 3 if the relative contribution of each component is known.

Yes, it would be useful to explicitly show the importance of each component in driving the trend of total PM2.5 in both observations and model simulations. The time series of the concentration share of each component (e.g. OC/Total PM2.5 %) is added in Fig. S6 in the supplement. However, the decomposition of the concentration share is not included since there is not much change in the percentage share in its trend component (few percentages at most in very limited cases) and the ratio does not necessarily have strong seasonality because of the phase difference in specific component and total PM2.5. Thus, including the trend component of time variant share of the ratio would only complicate the interpretation of the results. Instead, we have added a new Table 1 (see below) to show the overall concentration share of each component for both observations and model simulations to reflect the relative importance of different species.

5) In general, model evaluation is designed to improve model. It is difficult to relate the comparison results presented in this manuscript to specific model deficiencies in description of the chemical/physical processes and/or issues in model data sets, meteorological field and/or emission data. As sulphate, OC, nitrate are controlled by very different chemical processes, this reviewer would like to encourage the authors to further explore the difference in the comparison results for these species, which may reveal additional insights into the process-level model deficiencies.

We thank the reviewer for recognizing the potential of the proposed methodology in helping identify problems in the specific processes and/or model input. However, without running a new set of decadal simulations with a newer version of the model and/or modified temporal allocation of emissions, we cannot determine specific model deficiencies and/or issues in the model input data sets.

Specific Comments: 1) Figure 3 is hard to read because of log-log scale. It may be better to change the x-axis to the IMF number and y-axis to the ratio between model and observation characteristic scales. A second y-axis can be added to show the absolute characteristic scales for each IMF.

We thank the reviewer for the suggestion. However, because of the large discrepancies in the scales of IMFs (few days to thousands of days), log scale has to be employed to show the scales for all IMFs. Given that the characteristic periods are not easy to read from the plot, we provided the average characteristic periods for sub-seasonal and seasonal IMFs in the text. Moreover, since "not all IMFs from observation are being simulated and vice versa", a figure is needed for each site to show the characteristic scales (at least for the last few IMFs) separately for observations and model simulations. Thus, we have moved the inlet figures to Figure 3d-f for clarity and added the explanation in the caption. Adding a second y-axis and showing only observed characteristic scale would result in a very busy plot and we will not able to achieve the second point above. Please find our revision to the figure in the manuscript and below.

2) Section 4.2. Figure 6 shows some variation in time-derivatives. At the same, this reviewer would like to argue that about half of cases shown in the figure can be well approximated by linear assumption. The authors should comment on this aspect.

Linear assumption is useful in many cases, and linear trends do provide a general idea of magnitude of the change as well as whether the linear trend is significant or not. EMD is particularly useful for analyzing meteorological and pollutant time series, which are non-linear and non-stationary. The decomposed trend components can provide the exact time span and magnitude of a decreasing/increasing change throughout time. If we take the trend component of observed OC at ATL as an example, the OC level is stable at around 4.5 $\mu$g/m3 in 2002 and 2003 and decreases at varying rates during 2004-2007.

Please also note the supplement to this comment:

https://www.atmos-chem-phys-discuss.net/acp-2019-1079/acp-2019-1079-AC1-supplement.pdf

**Table 1.** Concentration share (%) of different components in total $PM_{2.5}$. It is estimated by dividing the mean trend components of each species by that of total $PM_{2.5}$ for both OBS and CMAQ, multiplied by 100. The concentration share of the remainder species (*Rem*) is estimated by subtracting all the available species share from 100 to compensate for the small discrepancies caused by the rounding up process and uncertainty in the mode decomposition. "-" indicates the data is not available (same applies for all other tables).

| | | $SO_4$ | $NO_3$ | $NH_4$ | OC | EC | Cl | Rem |
|---|---|---|---|---|---|---|---|---|
| QURE | OBS | 38 | 7 | - | 19 | 5 | 1 | 30 |
| | CMAQ | 19 | 15 | - | 14 | 5 | 1 | 47 |
| RENO | OBS | 7 | 13 | 5 | 46 | 11 | - | 20 |
| | CMAQ | 11 | 4 | 2 | 30 | 7 | - | 45 |
| ATL | OBS | 28 | 6 | 10 | 24 | 8 | - | 24 |
| | CMAQ | 22 | 10 | 8 | 17 | 9 | - | 33 |

**Fig. 1.**

[Figure]

**Fig. 3.** The characteristic scales ($t_m$) resolved in the IMFs of observed and simulated total and speciated $PM_{2.5}$ for (a, d) QURE, (b, e) RENO and (c, f) ATL. In (a-c), IMF1 to the last pair of IMFs with increasing characteristic periods are shown from bottom left to top right. Mean periods of IMFs with scales longer than a year are being displayed in (d-f) with the same shapes as in the legend above to show the characteristic scales of all decomposed IMFs given that not all IMFs from observation are being simulated and vice versa. In the (d-f), species decomposed from observations are shown with smaller filled shapes, while species decomposed from simulations are represented by larger open shapes in slightly darker shades.

**Fig. 2.**

---

## Author Comment (AC2) · 31 Jul 2020

Reply to interactive comments on "Evaluating Trends and Seasonality in Modeled PM2.5 Concentrations Using Empirical Mode Decomposition"

Anonymous Referee #4

General Comments: This paper introduces a new approach for process-based model evaluation of speciated PM2.5, which allows for the assessment of the performance of regional-scale air quality models like CMAQ on the intrinsic time-dependent longterm trend and cyclic variations in daily average PM2.5 and its species. The authors tested

the method with time series data at three sites. The data are generally sound, whereas some results and discussions of the study are still lack of persuasion.

One major concern is about how well the current approach's performance is compared with the previously published methods and some over-interpreted conclusions. The other is that it is not sure that the difference between the model and the new approach evaluation results can be simply explained by the inadequate description of nitrate or organics in the model. As the authors noted, they obtained abnormally low correlations of synoptic scale NO3 at ATL and calls for a better representation of nitrate partitioning and chemistry. What about the results for the other two sites? The authors need to provide more information on such issues to make the conclusion robust.

We appreciate the time and effort devoted by the reviewer to provide suggestions that helped improve the quality of our paper.

Our temporal decomposition approach applied to PM2.5 and its speciated components is not directly comparable with the other approaches reported in the literature. To avoid any over-interpretation of the analyses, we have refrained from exploiting model performance on the characteristic time scales and have carefully aligned our interpretation with IMFs that are statistically significant (almost all seasonal cycles are statistically significant from noise as shown in Fig. S5). Also, the differences between observed and simulated total and speciated PM2.5 are driven by several factors discussed in the paper. We cannot conclude exclusively that there is inadequate description of nitrate or organics in the model. Other potential issues such as the improper allocation of emissions also contributed to the difference between model simulations and observations. To be specific, description of secondary organic matter formation and magnitude and variation of primary sources are emerging areas of research; NO3 formation pathways are likely inadequately represented in the employed model version, and its predictions are also strongly influenced by the uncertainties in NH3 emissions.

CMAQ fails to simulate the magnitude of NO3 at all three sites with very abnormal

r_IMFn. Moreover, NO3 is the only component that has low correlation on the synoptic scale at ATL. The poor performance for NO3 mentioned above at all three sites calls for the modeler to look at the representation of nitrate partitioning and chemistry as summarized in the conclusions: "The consistent large under/over-prediction of NO3 variability at all temporal scales and magnitude in the trend component, as well as the abnormally low correlations on the synoptic scale NO3 at ATL, calls for better representation of nitrate partitioning and chemistry."

Specific comments:

1) Introduction: "Evaluation of ten-year averaged monthly mean of PM2.5 simulated with WRF/Chem ..." how does the model performance of PM2.5 compositions simulation should also be summarized to provide an intact view on the previous results.

Unfortunately, Yahya et al. (2016) only compared the overall 10-year average of the PM compositions (sulfate, ammonium, nitrate, EC, and total carbon) from ground-based observations to that of the model simulations as the background map. Thus, we are not able to make any conclusions on the seasonality of PM2.5 components.

2) Line 36: "and other natural species..." what do natural species refer to?

Natural species refer to PM2.5 non-anthropogenic components such as crustal material. We have changed this to "crustal elements" in the revised manuscript to avoid confusion.

3) Line 47: "monthly or seasonal means" means of speciated PM2.5?

The sentence is rephrased as: "monthly or seasonal means of total and/or speciated PM2.5."

4) Line 48: what do you mean by "ten-year averaged monthly mean"?

It is the monthly mean averaged over a period of ten years: ten-year averaged mean for Jan., Feb., ...

5) Line 51: "with a phase shift of few months" please explain phase shift.

The phase shift refers to that in Fig. 4c (copied below) in Yahya et al. (2016). The definition is similar to what we used in the evaluation of the cyclic signals: "the phase shift of an IMF n is defined as the days an IMF decomposed from modeled time series has to be shifted to maximize the correlation (R_max) with the corresponding IMF from observed PM2.5 time series."

6) Line 55-57: "...long-term trends or interannual variations driven by climate change, emission control policies or other slow varying processes..." what is the main reason? Are there any previous results?

Changes in air quality concentrations, such as PM2.5, are driven by changes in emissions and meteorological processes which highly impact the transport, chemical reactions and deposition of air pollutants. Thus, long-term trends reflect the impact of long-term changes in emissions (they might be governed by local control policies on anthropogenic emissions or climate-impacted natural emissions), long-term meteorological conditions (climate) and other slow varying processes (e.g. ENSO). There is no "main reason" among them. Here, we are simply stating that averaging over very long time periods can conceal signals driven by slow-changing processes: "In addition, averaging of those monthly or seasonal means across multiple years may conceal the long-term trends or interannual variations driven by climate change, emission control policies or other slow varying processes." We are not certain what the reviewer's query is directed at. Thus, we have left the sentence unaltered.

7) Line 68-74: I do not think this paragraph is necessary for the manuscript.

Following the reviewer's suggestion, the paragraph has been deleted in the revised manuscript.

8) Line 311: "RENO is located close to the border with California and is affected by large wildfire breakouts in the western U.S....." Is there any evidence for this demonstration?

The location of Reno, NV and the impact of California fire on July 10, 2008 is illustrated in Figure 1 (copied below) from Gyawali et al. (2009). We have also demonstrated the impact of 2008 fire season earlier in Section 4.1: "The small variation in the estimated characteristic period of IMF6 is because this monitoring site is located in a wildfire prone region on the border of Nevada and California. Clear evidence can be seen from Fig. 4a that an extra annual cycle in the IMF6 of observations in the summer of 2008 is depicted, which is very possibly driven by the 2008 California Wildfires spanning from May until November."

9) Line 327-: "To sum up, the long-term trend at QURE is well simulated by the model." This is unlikely consistent with the data presented in Table 1.

Our statement is based on the fact that the model has captured the decrease (i.e., rate of change), even though the absolute magnitude of the trend/long-term component is overestimated (which is what is shown in Table 1-now Table 2). We have re-phrased the sentence to: "To sum up, the decreasing long-term trend at QURE is well simulated by the model."

10) Lines 333-335: "Species other than those in the available dataset may also play a considerable role in driving the agreements or disagreements between model simulations and observations of total PM2.5" What are the contribution of these species to PM2.5 at the studied sites?

We have decomposed the remaining components (Rem) and added an 8th line of figures for the trend component in Rem in Fig. 6. The overall concentration share (%) of the remaining components can be found in the newly added Table 1. We have also added Figure S6 in the supplement that shows time series of the concentration share of each component (e.g. OC/Total PM2.5 %).

11) Lines 367-368: "Both observed and simulated annual cycles at the RENO site are

largely contaminated by the extreme events lasting for several months that are not properly simulated" is it possible to remove the data of extreme events before simulation, in order to eliminate the contamination?

These extreme events are very likely caused by large wildfires. We can eliminate emissions from wildfires in model simulations, but there is no straightforward way to eliminate contributions of wildfires in the observations. Thus, we kept the original observations and CMAQ model simulations, which included wildfire emissions.

12) Lines 384-387: "Specifically, the anti-correlation likely points to an inaccurate representation of the seasonal variation of the non-carbonaceous portion of organic matter due to an improper representation of organic aerosols in the model version analyzed here; this problem has since been corrected in more recent releases of the CMAQ model." This sentence needs to be rewritten for clearance. And what does the noncarbonaceous portion of organic matter refer to?

The long sentence has been revised for clarity: "Specifically, the anti-correlation likely points to an inaccurate representation of the seasonal variation of the non-carbonaceous portion of organic matter due to an incomplete representation of organic aerosols in the model version analyzed here; newer versions of the CMAQ model include updated treatment of organic aerosols (e.g., additional SOA formation pathways, improvements in representation of primary OM emissions) which is likely to correct the mentioned features (Appel et al., 2017; Murphy et al., 2017; Xu et al., 2018)."

The non-carbonaceous portion of organic matter refers to the portion of organic matter consisting of oxygen, hydrogen, and nitrogen.

Minor:

13) Line 17: "chloride (Cl) organic"

Corrected.

14) Line 311: "U.S. as can been seen"

Corrected.

Please also note the supplement to this comment:
https://www.atmos-chem-phys-discuss.net/acp-2019-1079/acp-2019-1079-AC2-
supplement.pdf

[Figure]

(c)

[Figure]

Fig. 4c in Yahya et al. (2016).

**Fig. 1.**

[Figure]

[Figure]

**Fig. 1.** Upper panel: Satellite image of smoke extending from northern California to Reno, Nevada on 10 July 2008. The smoke sources and wind trajectory were similar for much of July. Beneath panel: Conceptual model of emission and aging of urban and biomass burning aerosol.

**Fig. 2.**

**Table 1.** Concentration share (%) of different components in total PM$_{2.5}$. It is estimated by dividing the mean trend components of each species by that of total PM$_{2.5}$ for both OBS and CMAQ, multiplied by 100. The concentration share of the remainder species (*Rem*) is estimated by subtracting all the available species share from 100 to compensate for the small discrepancies caused by the rounding up process and uncertainty in the mode decomposition. "-" indicates the data is not available (same applies for all other tables).

| | | SO$_4$ | NO$_3$ | NH$_4$ | OC | EC | Cl | Rem |
|---|---|---|---|---|---|---|---|---|
| **QURE** | OBS | 38 | 7 | - | 19 | 5 | 1 | 30 |
| | CMAQ | 19 | 15 | - | 14 | 5 | 1 | 47 |
| **RENO** | OBS | 7 | 13 | 5 | 46 | 11 | - | 20 |
| | CMAQ | 11 | 4 | 2 | 30 | 7 | - | 45 |
| **ATL** | OBS | 28 | 6 | 10 | 24 | 8 | - | 24 |
| | CMAQ | 22 | 10 | 8 | 17 | 9 | - | 33 |

**Fig. 3.**